# Pathological Fracture of the Proximal Humerus Occurred on Metastases of Probable Kidney Origin in the Absence of Primary Lesions: A Case Report

**DOI:** 10.3390/healthcare11243108

**Published:** 2023-12-06

**Authors:** Luca Bianco Prevot, Stefania Fozzato, Luca Cannavò, Riccardo Accetta, Federico Amadei, Michela Basile, Massimiliano Leigheb, Giuseppe Basile

**Affiliations:** 1IRCCS Orthopaedic Institute Galeazzi, 20161 Milan, Italy; luca.bianco96@gmail.com (L.B.P.); basiletraumaforense@gmail.com (G.B.); 2Orthopaedic Department, Esine Hospital, 25040 Brescia, Italy; 3Hand and Peripheral Nerve Centre, COF Lanzo Hospital, 22020 Alta Valle Intelvi, Italy; 4Department of Biomedical and Dental Sciences and Morpho Functional Imaging, University of Messina, 98122 Messina, Italy; 5Orthopaedics and Traumatology Unit, “Maggiore Della Carità” Hospital, Department of Health Sciences, University of Piemonte Orientale (UPO), Via Solaroli 17, 28100 Novara, Italy

**Keywords:** pathological fracture, cancer of unknown primary site (CUP), immunohistochemistry, metastasis, treatment, proximal humerus

## Abstract

Cancer of unknown primary (CUP) origin represents a diagnostic and therapeutic challenge. These tumours spread to different parts of the body even if the site of origin has not been identified. When renal metastases are observed without an obvious primary lesion, it is important to exclude the possibility of a primary kidney tumour that may be unknown or too small to be detected. The diagnosis of CUP is established after a careful clinical evaluation and diagnostic tests, including blood chemistry and laboratory tests, instrumental exams (CT, MRI, PET, bone scan), biopsy, and molecular and cytogenetic analysis. Once the diagnosis of CUP with kidney metastases is confirmed, treatment depends on the location of the metastases, the patient’s health status, and available treatment options. The latter includes surgery to remove metastases, radiation therapy, or systemic treatment such as chemotherapy or immunotherapy. It is important that patients with CUP are evaluated by a multidisciplinary team of specialists, who can contribute to planning the most appropriate treatment. In this article, we report the clinical case of a patient with a pathological fracture of the proximal humerus which occurred on metastases of probable renal origin in the absence of primary lesions.

## 1. Introduction

Metastases represent the dissemination of cells from a primary tumour to other distant organs or tissues and constitute one of the evolutionary phenomena of oncological pathology with the worst prognosis. In fact, approximately 90% of registered deaths in oncologic patients are related to metastatic complications [1]. Metastases may represent the only clinical manifestation of occult or undetectable primary malignancies. Tumours of unknown origin (CUP) constitute approximately 2–4% of all new cancer cases, with an annual incidence of 6–12 cases per 100,000 inhabitants [2]. In patients with CUP referred for feedback at autopsy, a primary renal site was identified in approximately 5% of cases. However, in the literature, only a few cases of patients with CUP have been diagnosed with renal cell carcinoma (RCC) [3]. The timely diagnosis of primary renal tumour is of fundamental importance, since today several chemotherapy drugs are available, such as immune checkpoint blockers, which significantly improved survival expectations even in advanced stage tumours [4]. A key aspect is that some diagnoses of CUP are premature or even completely incorrect and the diagnosis of CUP remains valid if no primary tumour is detectable in the respective organ. The correct diagnosis of CUP also strongly depends on the clinical judgment and experience of the treating oncologist [5]. When CUP is suspected, close interdisciplinary collaboration between pathology, medical oncology, and imaging is essential to obtain a valid classification similar to CUP or the identification of a presumed primary tumour, in the interest of providing the most specific and effective therapy for the patient [6].

## 2. Case Presentation

A 62-year-old male patient was taken to the emergency room of the Scientific Institute for Research, Hospitalization and Health Care (IRCCS) Galeazzi Orthopaedic Institute (Milan, Italy) for pain and functional impotence in the right shoulder following an accidental fall at home with direct trauma to the upper limb. During the anamnestic collection, the patient reported being affected by arterial hypertension and hypercholesterolemia. He also reported having suffered a previous ST-elevation myocardial infarction (STEMI), treated with drug-eluting stent implantation. The patient reported episodes of fever, weight loss, and asthenia in the year preceding the traumatic event. A shoulder radiograph with standard projections was performed, which allowed for the diagnosis of a humeral surgical neck fracture, classifiable as type 11-A2.2 according to the AO/OTA Fracture and Dislocation Classification revised in 2018 (Figure 1A). The results of routine blood tests performed on admission were normal. Two days after the traumatic event, the patient underwent surgery to reduce and fix the fracture with a T2 ^®^ model 8 mm × 150 mm intramedullary nail and screws (Stryker, Kalamazoo, MI, USA) (Figure 1B) under general anaesthesia. The immediate postoperative course was regular, with good control of painful symptoms and reported subjective well-being. On the second postoperative day, the patient was discharged with indications for clinical follow-up (FU) and periodic radiographs.

The evolution of the patient’s clinical picture upon returning home was partly monitored with telemedicine [7] through the installation of an application on a mobile device which allowed direct and continuous communication with healthcare professionals, thus allowing personalised treatment, intensification of follow-ups, and adequate continuity of care, while simplifying the patient’s care relationship. At the 1-month FU, the patient reported subjective well-being, good control of blood pressure, and absence of secondary displacements on radiographic control. For this reason, an adequate rehabilitation program was planned aimed at recovering joint function and gradually restoring physiological limb proprioception. At the FU approximately two months after the operation, the patient complained of worsening pain in the operated limb and functional limitation of the shoulder. The operated area appeared swollen, oedematous, and red. The shoulder radiograph highlighted an extensive area of humeral metadiaphyseal osteolysis (Figure 2A), not present in the previous instrumental images. For this reason, it was decided to perform a CT examination of the shoulder, which confirmed the presence of an osteolytic lesion associated with the presence of pathological tissue at the level of the periskeletal system on the postero-medial humeral side (Figure 2B).

Given the suspicion of a neoplastic lesion, it was decided to carry out an in-depth diagnostic with an MRI examination to complete the local instrumental imaging which highlighted a subversion of a voluminous portion of tissue in the proximal III of the humerus with manifestation in the periskeletal soft tissues, compatible with a replacement lesion (Figure 3). S CT-guided biopsy was performed to realise histological examination.

The outcome of the biopsy identified the lesion as a “bone metastasis of carcinoma with extensive areas of necrosis and sarcomatoid features, compatible with metastasis from renal cell carcinoma (immunohistochemistry: CD10+, Vimentin+, PAX8+, CKAE1/AE3+, GATA3-, CK7-, CK10-, TTF1-, p40-, p63-)”. To allow for the formulation of a correct oncological staging, the patient underwent further investigations such as cranial-brain, chest, and abdominal CT, which did not reveal suspicious findings for primary neoplastic lesions. Fluorodeoxyglucose positron emission tomography did not detect any malignant lesions. At the end of the investigations, the patient underwent surgery to resection the proximal two-thirds of the humerus (Figure 4A) and implant a mega prosthesis (Figure 4B).

After the surgical treatment with a mega prosthesis, the patient was transferred and treated in a specialised oncology department in another hospital. 

The histological examination performed on the surgical specimen confirmed the biopsy diagnosis, identifying the lesion as a metastasis from renal cell carcinoma. 

At the orthopaedic FU, one month after the mega prosthesis implant surgery, instrumental images showed that the implant was well positioned. At the same time, the patient presented a eutrophic scar, with a range of motion (ROM) of the shoulder of 15° in abduction and flexion and an external rotation of 20°.

The patient underwent a cycle of adjuvant radiotherapy in the arm affected by the localisation of the metastasis at the oncology department.

Diagnostic tests, such as genetic studies and DNA analyses, are still underway to identify the neoplastic lesion and evaluate the opportunity for chemotherapy treatment.

## 3. Discussion

Metastatic tumours of unknown primary origin (CUP) are malignancies that manifest as metastatic lesions in the absence of a primary tumour. Patients with CUP who have an occult primary renal site are rare, representing approximately 5%. Standard empiric chemotherapy administered in cases of CUP with regimens of platinum and taxane, gemcitabine and platinum, or fluoropyrimidine is often ineffective in cases of RCC. However, the treatment of advanced RCC has significantly improved in recent years with the introduction of more latest-generation pharmacological molecules, such as immune checkpoint blockers [8].

Our proposed case report represents an interesting case of fracture of the proximal humerus without pathological signs, which proved to be an evolutionary lesion compatible with renal tumour metastasis in the absence of focal lesions to the kidneys. Patients with metastatic tumours of unknown primary origin that present with bony lesions tend to have a lower survival expectancy than patients with other presentations of CUP. Difficulties with tissue acquisition and analysis play a significant role in diagnosis and can delay treatment. Current routine decalcification protocols for processing bone samples damage nucleic acids, leading to a high failure rate for profiling the molecular structure of the lesion [9].

Most patients with bone metastases with CUP manifest with pathological fractures and require local treatment which can potentially delay the initiation of any systemic therapy, resulting in further uncontrolled progression of systemic disease, as occurred in our case [10]. In our case, the suspicion of a secondary lesion starting from a renal tumour was raised following the histological examination and immunohistochemical investigations which were performed both on the tissue coming from the biopsy and on the resected mass. 

Immunohistochemical investigations are of fundamental importance for diagnosis and to identify possible therapeutic targets, allowing innovative paths to be undertaken through ethically, legally, and socially valid approaches, aimed at improving the patient’s quality of life. This would allow us to undertake personalised medicine which, as already known [11], is by its nature information-intensive and of which the predictive, diagnostic, and therapeutic capabilities are based on high-dimensional data, corroborated by a set of multidisciplinary analytical skills. Furthermore, a great innovation in this field has been provided by artificial intelligence (AI), which has recently been progressively used in many fields of medicine [12], integrating knowledge and information technology through machine learning algorithms, resulting in an improvement in treatment options and more optimal planning of procedures. 

Since CUP represents a complex and unique clinical situation for each patient, it is essential to involve a multidisciplinary team of specialists, including oncologists, radiologists, pathologists, and surgeons, to develop a personalised treatment plan based on the specifics of the case. Symptoms of metastatic tumours of unknown primary origin in the kidney may vary depending on the parts of the body where the metastases have spread. Since there is no identifiable primary lesion in the kidney, symptoms can be attributed to the metastases themselves and may include pain at the metastatic site, asthenia, weight loss, loss of appetite, gastrointestinal disorders (nausea, vomiting, abdominal pain), respiratory disorders (persistent cough, difficulty breathing or chest pain), and neurological disorders. Since the unknown origin is unfortunately related to a short life expectancy, chemo-radiotherapy and surgery usually only have a palliative role [13], even more so in immunosuppressed subjects where the lack of adequate therapies makes it extremely difficult to develop adequate treatment strategies, prevention, and treatment [14].

Spontaneous tumour regressions attributed to the elimination of tumour cells by the immune system have been reported. The concept of immune-mediated cancer surveillance is further supported by an increased incidence of cancer in transplant patients taking immunosuppressants and the recent success of immune checkpoint inhibitors in numerous tumour entities. Therefore, it is possible that the lack of a primary tumour in CUP is an immune-mediated event, at least in some cases [15]. However, for definitive classification as CUP, a comparison with medical oncological and imaging results is essential.

A multidisciplinary approach helps provide a more complete and accurate assessment of the case, allowing the most appropriate and personalised treatment plan to be defined for the patient. A multidisciplinary approach involving various specialists in the diagnosis of tumours of unknown primary renal origin (CUP) offers several significant advantages, which include more accurate evaluation, better therapeutic decisions, more effective management of complications, and active patient involvement, through clear communication, sharing decisions, discussion of the patient’s values and preferences, and shared decision support, with the aim of improving the quality of care. However, it is important to be aware of the possible medico-legal implications that could arise in relation to this type of diagnostic–therapeutic approach.

In the end, we believe that multicentre studies are first necessary to propose algorithms that should be based on technologies commonly used in all hospitals, to help in common clinical practice. Furthermore, the primary aim of this work is not to provide diagnosis or treatment algorithms which must necessarily be personalised to the patient, but to remind the reader that sometimes a much more serious pathology can hide behind the most banal case.

## 4. Conclusions

In conclusion, we can state that patients with metastatic bone lesions from an unknown primary tumour represent a therapeutic challenge for the orthopaedic surgeon, who is generally the first doctor encountering the patient, very often following a pathological fracture. It is therefore the healthcare worker who must not only carry out local management and treatment of the disease, but also activate a multidisciplinary path for a correct diagnosis and adequate oncological therapy. In these cases, a multidisciplinary treatment must always be pursued for adequate local and systemic control of the tumour.

## Figures and Tables

**Figure 1 healthcare-11-03108-f001:**
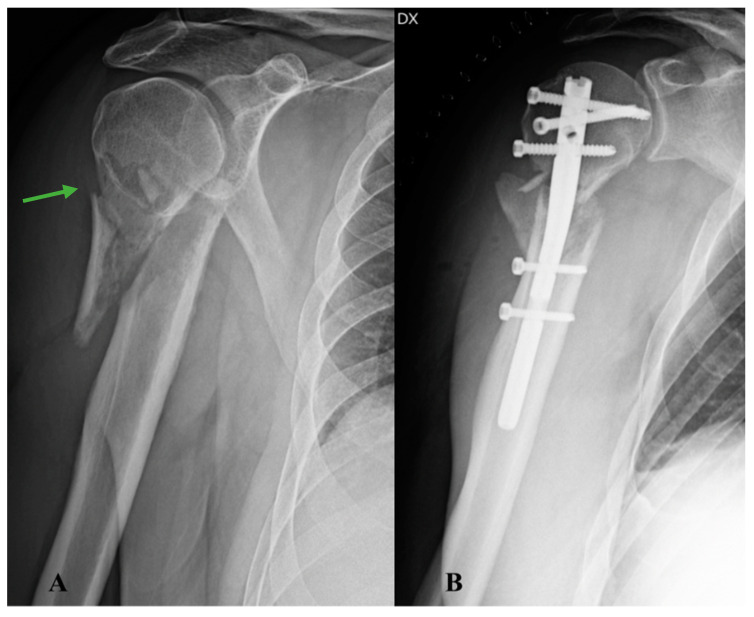
Right shoulder X-rays: (**A**) anteroposterior radiograph of the shoulder showing a surgical neck fracture of the proximal humerus (arrow); (**B**) antero-posterior radiograph of the postoperative shoulder showing reduction of the major fracture fragments and fixation with T2 ^®^ model 8 mm × 150 mm intramedullary nail and screws (Stryker, Kalamazoo, MI, USA).

**Figure 2 healthcare-11-03108-f002:**
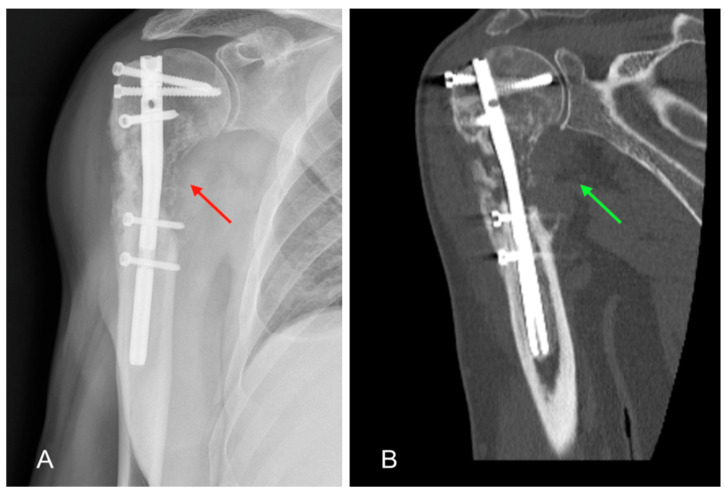
X-ray and CT scan at 2-month FU: (**A**) antero-posterior radiograph of the shoulder showing an area of extensive humeral metadiaphyseal osteolysis (red arrow); (**B**) CT image of the shoulder in coronal section documenting the presence of pathological tissue at the periskeletal level in the postero-medial aspect of the arm (green arrow).

**Figure 3 healthcare-11-03108-f003:**
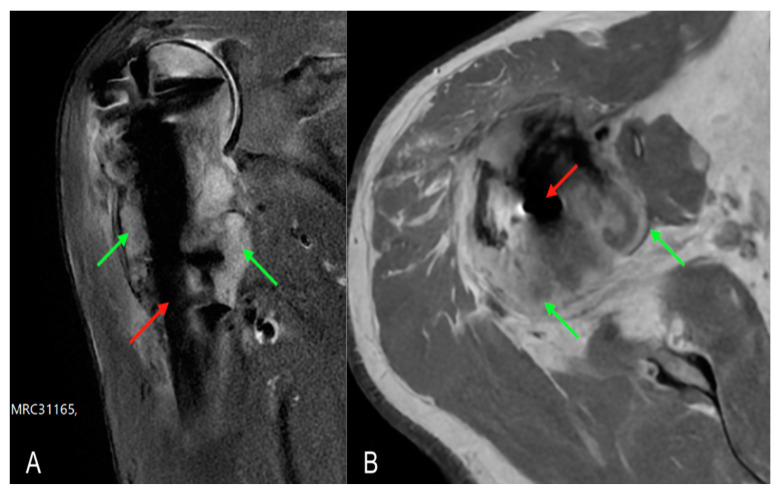
MRI scans of the left arm showing intramedullary nail (red arrow) and periskeletal neoplastic pathological tissue (green arrows) at the level of the humeral head and diaphysis: (**A**) coronal MRI section; (**B**) transversal MRI section.

**Figure 4 healthcare-11-03108-f004:**
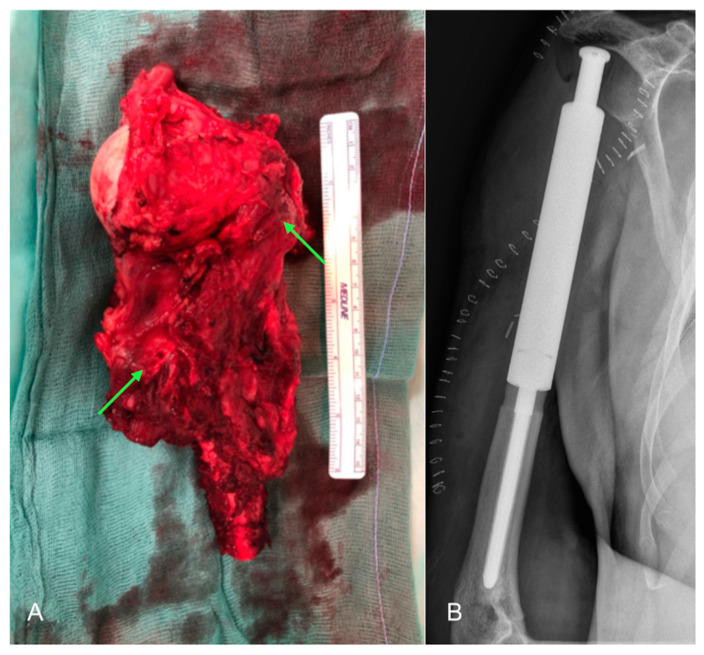
Intraoperative image and post-operative X-ray: (**A**) Resection of the proximal two thirds of the left humerus with a neoplastic mass surrounding the humeral head and part of the diaphysis (green arrows); (**B**) postoperative antero-posterior X-ray of the left arm showing the implanted mega prosthesis.

## Data Availability

All data analysed during the current study are available from the corresponding author upon reasonable request.

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
