# Peer review of "Pathological Fracture of the Proximal Humerus Occurred on Metastases of Probable Kidney Origin in the Absence of Primary Lesions: A Case Report"

_healthcare, 2023, doi:10.3390/healthcare11243108_

Round 1
Reviewer 1 Report
Comments and Suggestions for Authors
In this manuscript, the Authors described the case report of a patient with a pathological fracture of the proximal humerus which can be explained as a metastasis of probable renal origin in the absence of primary lesions.
The case report is well written. However, the patient has not started the treatment yet and, consequently, an essential part of the clinical history is missing (the response to the treatment). I suggest resubmitting the case report after the first radiological re-assessment.
Furthermore, I suggest adding other details in the case description (if available): i) Has the patient undergone a PET CT? ii) What are the values of tumoral markers? iii) Has the patient undergone endoscopic exams (colonoscopy, gastroscopy)?
In addition, are there any similar cases already published in the literature? If yes, I suggest reporting them in the "Discussion" paragraph.
Finally, chemotherapy is not the standard treatment for kidney cancer (Line 127-128).
Comments on the Quality of English Language
Minor editing of English language required
Reviewer 2 Report
Comments and Suggestions for Authors
Cancer of unknown primary site (CUP) is a relatively common clinical entity. Within this category, tumors from many primary sites with varying biology are represented. This heterogeneity has made the design and interpretation of clinical studies difficult.
This is a case report of humeral metastasis of CUP, compatible with renal carcinoma, but CT negative in that area.
The text is well written, it is easy to read, and to understand.
Comments:
(1) Line 36, regarding "In fact, approximately 90% of registered deaths are related to metastatic complications". Of registered deaths of cancer patients?
(2) Line 30, could you please add "site" after "primary"?
(3) Line 43, there are several histological subtypes of primary renal tumor. Could you please describe the most frequent and the ones with more capacity to produce metatases?
(4) Line 55. Could you please add "Scientific Institute for Research, Hospitalization and Health Care" before "IRCCS"?
(5) Line 58, "arterial hypercholesterolemia"?
(6) Line 59, do you mean Acute ischemic stroke (AIS) is a rare following ST-elevation myocardial infarction (STEMI)?
(7) Line 63. Are you using the new AO/OTA 2018 classification? Is it the A2.1 simple surgical neck fracture?
(8) In Figure 1, the fracture is easily visible, but may you please add an arrow?
(9) Line 107. (9.1). Could you please show the histological images of the renal cell carcinoma?
9.2. Do you have access to the hematoxylin and eosin stains?
9.3. Do you have access to the immunohistochemical stains?
9.4. There are several histological subtypes, for example chromophobe, clear cell, oncocytoma, papillary, translocation, SDH deficient RCC. Please confirm.
9.5. What was the differential diagnosis made by the pathologist? Please expand.
(10) As I understand, the abdominal CT scan was negative? What about the kidneys? What diagnostic tests are underway?
(11) Discussion, lines 130 to 190. This is a single paragraph. Could you please divide it into different parts?
Reviewer 3 Report
Comments and Suggestions for Authors
Indeed, cancer of unknown primary (CUP) poses a diagnostic and therapeutic challenge. These tumors spread to different parts of the body, even if the site of origin is not known. The authors consider a clinical case of a patient with a pathological fracture of the proximal humerus, which occurred against the background of metastases of probable renal origin in the absence of a primary lesion. The diagnosis of CUP is made after a thorough clinical examination and diagnostic studies, including biochemical blood tests and laboratory tests, instrumental studies (CT, MRI, PET, bone scan), biopsy, molecular and cytogenetic tests. The authors show that therapy for CUP includes surgery to remove metastases, radiation therapy, or systemic treatment such as chemotherapy or immunotherapy. It is important that patients with CUP are assessed by a multidisciplinary team who can contribute to planning the most appropriate treatment.
The article is well written, well illustrated and undoubtedly deserves the attention of readers. However, it seems to me that in the discussion section it would be appropriate to provide a scheme (sequence) for diagnosing CUP and choosing a treatment algorithm, if possible.
Round 2
Reviewer 1 Report
Comments and Suggestions for Authors
The Authors have addressed the suggestions and comments listed in my previous report. However, an extensive english language editing is required.
Comments on the Quality of English LanguageAn extensive english language editing is required.